# Experimental Study of the Influence of Selected Factors on the Particle Board Ignition by Radiant Heat Flux

**DOI:** 10.3390/polym14091648

**Published:** 2022-04-19

**Authors:** Ivana Tureková, Martina Ivanovičová, Jozef Harangózo, Stanislava Gašpercová, Iveta Marková

**Affiliations:** 1Department of Technology and Information Technologies, Faculty of Education, Constantine the Philosopher University in Nitra, Tr. A. Hlinku 1, 949 74 Nitra, Slovakia; ivana.turekova@ukf.sk (I.T.); mabumb@gmail.com (M.I.); jozef.harangozo@ukf.sk (J.H.); 2Department of Fire Engineering, Faculty of Security Engineering, University of Žilina, Univerzitná 1, 010 26 Zilina, Slovakia; stanislava.gaspercova@uniza.sk

**Keywords:** ignition, particleboard, radiant heat, thermal resistance, ANOVA

## Abstract

Particleboards are used in the manufacturing of furniture and are often part of the interior of buildings. In the event of a fire, particleboards are a substantial part of the fuel in many building fires. The aim of the article is to monitor the effect of radiant heat on the surface of particle board according to the modified procedure ISO 5657: 1997. The significance of the influence of heat flux density and particle board properties on its thermal resistance (time to ignition) was monitored. Experimental samples were used particle board without surface treatment, with thicknesses of 12, 15, and 18 mm. The samples were exposed to a heat flux from 40 to 50 kW·m^−2^. The experimental results are the initiation characteristics such as of the ignition temperature and the weight loss. The determined factors influencing the time to ignition and weight loss were the thickness and density of the plate material, the density of the radiant heat flux and the distance of the particle board from the radiant source (20, 40, and 60 mm). The obtained results show a significant dependence of the time to ignition on the thickness of the sample and on the heat flux density. The weight loss is significantly dependent on the thickness of the particle board. Monitoring the influence of time to ignition from sample distance confirmed a statistically significant dependence. As the distance of the sample from the source increased, the time to ignition decreased linearly. As the distance of the sample from the source increased, the time to ignition increased.

## 1. Introduction

Sheet board materials are among the most important wood products [1]. Their production encompasses utilization of wood of lower quality classes and obtaining suitable materials with improved physical and mechanical properties [2].

This product group contains wood-based boards for the use in building interiors, such as boards without surface treatment (raw) or with surface treatment (particleboards), plywood, fiberboard, and edge-glued wood panels [3,4].

Particleboard can be defined according to STN EN 309:2005 [4] as a molded wood material, produced by heat pressing of small wood particles (e.g., chips, shavings, sawdust, lamellas, etc.) or other lignocellulosic particles (e.g., flax shives, hemp shives, bagasse, etc.) with adhesives.

The processing wood of all woody plants occurring in Central Europe is used as a source of wood in the production of particleboards. These are less valuable forest assortments, industrial and residual waste, recycled wood, and other lignocellulosic materials [5].

Particleboards belong to a product group of board materials, but they are considered an input material in the furniture and construction industries [6]. In terms of quality assessment, particleboards have only few disadvantages, and flammability is among them [7,8,9,10,11].

The current state of technology and production techniques in particleboard production allow processing of practically all types of wood occurring in Central Europe using a suitable mixture [12,13,14].

Wood and sheet board materials represent a substantial part of the fuel in many building fires [15].

The assumption of a fire hazard requires an appropriate description of the fire ignition and fire development [16,17]. The initial process is ignition [18]. Flammability can be defined as the ability of materials to ignite when heated to elevated temperatures. It depends on many factors, mainly the critical heat flux and the thermal properties of materials. Currently, there are several methods for determining the flammability, fire-technical, and physical material properties, which are defined by relevant standards [19,20].

The aim of this article is to analyze the influence of heat flux density and particle board properties (thickness of 12 mm, 15 mm, 18 mm and board material density) on their thermal resistance (time to ignition) and ignition characteristics (ignition temperatures and weight loss). This dependence was also monitored when the distance of the sample from the radiant heat source changed, which represents an important safety factor in the ignition phase of real fires.

## 2. Materials and Methods

### 2.1. Samples

Particleboard research is part of improving their properties [21,22,23,24]. Separate attention is paid to the research of the physical and mechanical properties of the particleboards [25,26,27,28].

Particleboards with thicknesses of 12, 15, and 18 mm were used for experiments due to their practical applicability and popularity in practice (Figure 1). Selected materials are among the most widely used materials nowadays in the furniture and construction industry [29,30]. Particleboard samples were sourced from the company BUČINA DDD, Zvolen, Slovakia [31,32] under product name Particleboard raw unsanded (Table 1). Particleboards contain coniferous softwood chips, mainly spruce and urea-formaldehyde adhesive mixture.

Large-size wood materials form the largest percentage of wood material in timber houses which means they can be directly exposed to fire [33].

Selected thicknesses of board materials are used in the construction and insulation of houses, in the construction of ceilings, soffits, partitions, etc.

**Figure 1 polymers-14-01648-f001:**
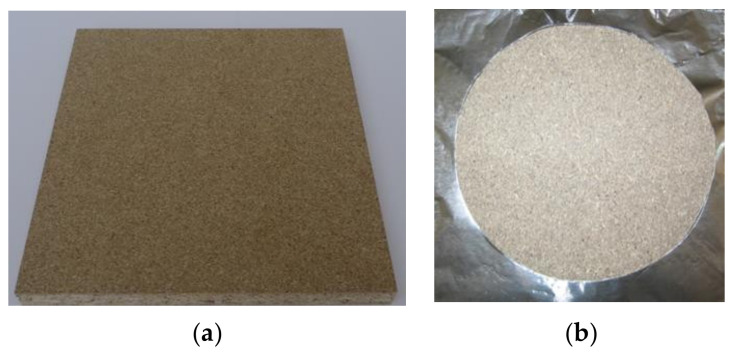
Example of experimental samples. (**a**) Particleboard; (**b**) sample prepared for measurements in accordance with ISO 5657 [34].

Particleboard samples were cut to specific dimensions (165 × 165) mm according to STN 5657: 1997 [34]. Selected board materials were kept at a specific temperature (23 °C ± 2 °C) and relative humidity (50 ± 5%).

### 2.2. Methodology

The density of the particleboards was determined according to STN EN 323: 1996 [35]. The time to ignition and weight loss depending on the selected level of heat flux density and thickness of board materials and the distance of selected board materials from the ignition source was determined according to the modified procedure ISO 5657: 1997 [34]. A detailed description of the modification and the course of the experiment is described in Tureková et al. [36].

The heating cone ensures heat flow in the range of 10 to 70 kW·m^−2^. The heat acts in the center of the hole in the masking plate where the test sample is placed (Figure 2).

The heating cone temperatures were verified by a thermocouple that is in close and constant contact with the heating element tube, and the heat fluxes were determined on the basis of a calibration curve [36].

The samples were placed horizontally and exposed to a heat flux of 43 to 50 kW·m^−2^ by an electrically heated conical radiator. Orientation experiments determined the minimum heat flux required to maintain flame combustion.

The horizontally placed sample under the thermal cone is exposed to the selected heat flux and gradually thermally degrades. During the experiment, the course of degradation is monitored, which is manifested by weight loss. At the same time, time to ignition is monitored. Time to ignition was recordedwhile considering only the permanent ignition of the surface of the analyzed sample when exposed to a selected level of heat flux density.

Thermal inertia, which is closely related to the time to ignition, was calculated for each selected board material [37]. The higher the thermal inertia value, the slower the temperature rise on the surface of the board material and the later the ignition [38,39,40]. Thermal inertia was calculated according to Schieldge et al. [41]:I = λ ·ρ· c·[J^2^·m^−4^·s^−1^·K^−2^](1)
where λ [W·m^−1^·K^−1^] is the thermal conductivity, ρ [kg·m^−3^] is the board material density, and c [J·kg^−1^·K^−1^] is heat capacity.

The influence of the ignition source distance on the time to ignition of the board materials was monitored on particleboards with a thickness of 12 mm. The choice of thickness was made from a practical point of view. Particleboards with a thickness of 12 mm are the most commonly used materials in the construction industry in thermal insulation, timber houses, construction of ceilings and soffits [42].

The experiments were performed with the radiant heat fluxes of 44, 46, 48, and 50 kW·m^−2^. The distance between the cone calorimeter and the particleboard was 20, 40, and 60 mm. The choice of distance was determined based on orientation experiments and changes in times to ignition were monitored even in case of minimal changes in distance from the ignition source. A preparation consisting of cement cubes measuring 20 × 20 × 20 mm was used to change the distance of the board material from the ignition source. The experiments were repeated five times.

Specific factors affecting time to ignition and weight loss are:Thickness and density of the board material;Radiant heat flux density;Distance of particleboards from the radiant heat source.

### 2.3. Mathematical and Statistical Processing of Data and Evaluation of Results

To evaluate the influence of the above-mentioned factors on the ignition temperature and weight loss, the obtained results were subjected to a statistical analysis. The obtained results of the ignition and weight loss temperatures were statistically evaluated by two-way analysis of variance (ANOVA) using the least significant difference (LSD) test (95%, 99% detectability level), (STATGRAPHICS software version 18/19 (Statgraphics Technologies, Inc., The Plains, VA, USA), with the following influence factors: board material thickness (12, 15 and 18 mm), radiant heat flux density (from 43 to 50 kW·m^−2^), and distance of board materials from the ignition source (20, 40 a 60 mm).

## 3. Results and Discussion

The course of the experiment (Figure 3) according to ISO 5657: 1997 [34] confirmed the verified behavior of the material in terms of the classification “reaction to fire (D-s1, d0)” [43,44,45] (Figure 4). The priority of the experiment is to monitor the critical parameters of the ignition based on the change in board thickness (Figure 3 and Table 2).

### 3.1. Determination of Ignition Temperature and Weight Loss

The ability of the material surface to generate volatile gases when exposed to radiant heat as well as the ability of selected board materials to ignite when exposed to radiant heat fluxes caused by an ignition source were confirmed.

The density of samples ranged from 640 to 720 kg·m^−3^. This range corresponds to the usual density of particleboards [46].

By comparing the calculated thermal inertia with the reported thermal inertia by Babrauskas [47,48], very similar results were confirmed. The thermal inertia values ranged from 0.31 to 0.33 kJ^2^·m^−4^·s^−1^·K^−2^. The difference was around 0.02 kJ^2^·m^−4^·s^−1^·K^−2^ in specific particleboards.

In this case, it is not possible to look for the dependence of inertia on other parameters, as the ANOVA results show in Table 3.

Figure 5 shows a statistically significant dependence of the time to ignition on the sample thickness. This dependence was made for heat fluxes of 43–48 kW·m^−2^.

The ANOVA table (Table 3) decomposes the variability of Col_4 (Time to ignition) into contributions due to various factors. Since Type III sums of squares (the default) have been chosen, the contribution of each factor is measured having removed the effects of all other factors. The *p*-values test the statistical significance of each of the factors. Since one *p*-value is less than 0.05, this factor has a statistically significant effect on Col_4 at the 95.0% confidence level. Dependence of the decrease in time to ignition on the increase in heat flux and the increase in the particleboard thickness was confirmed (Figure 6).

These dependencies are statistically significant (Table 4). The *p*-values test the statistical significance of each of the factors. Since 2 *p*-values are less than 0.05, these factors have a statistically significant effect on Col_3 at the 95.0% confidence level (Figure 7).

Figure 8 describes the weight loss in selected particleboard thicknesses when exposed to radiant heat flux (40–50 kW·m^−2^). As the heat flux density increases, the value of the weight loss in the particleboard samples of the selected thicknesses increases on average by 0.4% (absolute % number) for a change of the heat flux of 1 kW·m^−2^. The largest weight loss values were recorded in particleboards with a thickness of 12 mm.

The course of the increase in weight loss as a function of increasing radiant heat flux is statistically significant. This statement is based on a statistical analysis of the STATGRAPHICS Software Program version 18/19. The ANOVA method was used (Table 5, Figure 8), where the *p*-values test the statistical significance of each of the factors. Since 2 *p*-values are less than 0.05, these factors have a statistically significant effect on Col_3 at the 95.0% confidence level (ANOVA). confidence level (Figure 9).

Valcheva and Savov [49] also presented scientific experiments covering characteristic features and the effect of different thicknesses of boards. The regression models describing the effect of thicknesses on main properties of medium-density particleboard are deduced and analyzed from the output data.

### 3.2. Monitoring the Effect of the Distance of Board Material from the Ignition Source

The distance of the particleboard from the radiant heat source (Figure 10 and Figure 11) has influence on the time to ignition (Table 5). Particle boards were ignited at higher heat fluxes from 44 kW·m^−2^ at a distance of 40 mm (Figure 10a) particleboards were ignited only if the heat flux was at least 44 kW·m^−2^; for 60 mm the lowest heat flux for ignition was 48 kW·m^−2^ (Figure 10d). The higher the heat flux, the shorter the time to ignition. Particleboards accumulated sufficient heat to allow the subsequent combustion without the action of an ignition source on the upper surface of the board material.

The obtained time to ignition has a decreasing character with a linear dependence. At the distance of 60 mm and heat fluxes of 44 and 46 kW·m^−2^, ignition did not occur. However, the imaginary line through two points showing the ignition temperature values at the distance of 60 mm shows a different tendency. Ignition temperatures doubled. It can be assumed that with the increasing distance of the radiant heat source from the sample, the increase in time to ignition multiplies geometrically. Time to ignition is significantly dependent on the heat flux and sample thickness (Table 6, Figure 12), as confirmed by multifactor analysis (ANOVA).

Statistical analysis demonstrated a significant influence of factors such as distance of the heat-stressed sample (Figure 11) and heat flux (Figure 6) on the time to ignition under the action of radiant heat flux on the surface of particleboards.

Also, statistical analysis ANOVA showed effect of distance on time to ignition (Figure 12). The abbreviations Col_4–distance and Col_6–time to ignition, were used in the graphical representation of statistical results (Figure 12).

## 4. Conclusions

Based on the conducted experiments, the following results were obtained:It was statistically confirmed that the time to ignition is significantly dependent on the thickness of the particleboard sample and the heat flux value. It was also possible to calculate the thermal inertia based on the measurements. The obtained results of the calculated inertia were very similar to the published values reported by Babrauskas [48];It was confirmed that the weight loss was significantly dependent on the thickness of the particleboard. Selected thicknesses of particleboards which were exposed to radiant heat flux of 40–50 kW·m^−2^ recorded on average by 0.4% (absolute % number) of weight loss with increasing heat flux density (for a change of the heat flux of 1 kW·m^−2^). The largest weight loss values were recorded in particleboards with a thickness of 12 mm;Statistically significant dependence was confirmed by monitoring the time to ignition and the distance of a sample with a thickness of 12 mm from the radiant heat source. At a distance of 60 mm and heat fluxes of 44 and 46 kW·m^−2^, the particleboards with a thickness of 12 mm did not ignite.

## Figures and Tables

**Figure 2 polymers-14-01648-f002:**
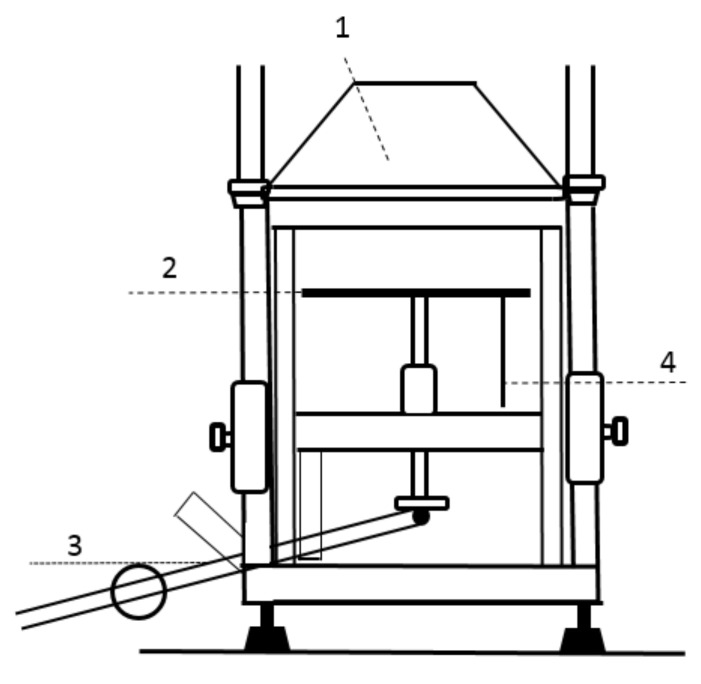
Scheme of the equipment for determination of flammability of materials at a heat flux of radiant heat of 10–50 kW·m^−2^ according to ISO 5657: 1997 [34]. Legend: 1-heating cone, 2-board for sample, 3-movable arm, 4-connection point for recording experimental data.

**Figure 3 polymers-14-01648-f003:**
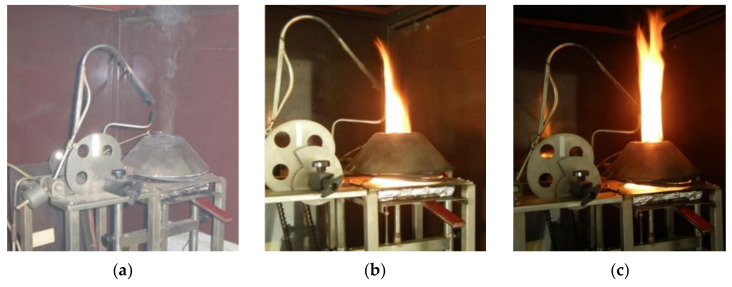
Course of measurement of time to ignition and weight loss for a particleboard sample with a thickness of 15 mm, heat flux intensity 45 kW·m^−2^. Legend: (**a**) sample ignition (time to ignition 84 s); (**b**) burning of the sample in 100 s; and (**c**) burning of the sample in 120 s.

**Figure 4 polymers-14-01648-f004:**
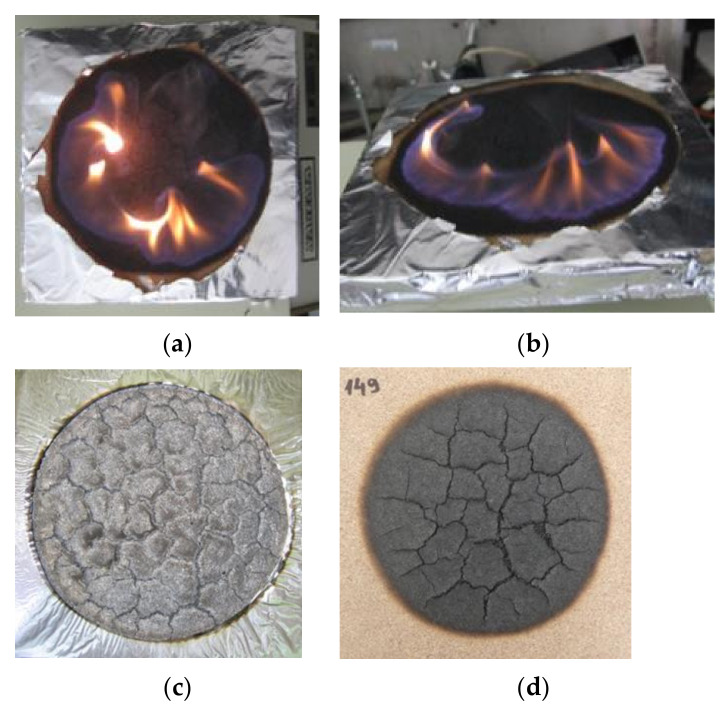
Combustion process of particleboards after their ignition by radiant heat (**a**) top view for sample with 15 mm thickness immediately after experiment; (**b**) side view for sample with 15 mm thickness immediately after ignition; (**c**) sample with 15 mm thickness taken out from the measuring device, placed at a distance of 20 mm after the end of the experiment; (**d**) cooled sample 10 min after the experiment, sample thickness of 18 mm.

**Figure 5 polymers-14-01648-f005:**
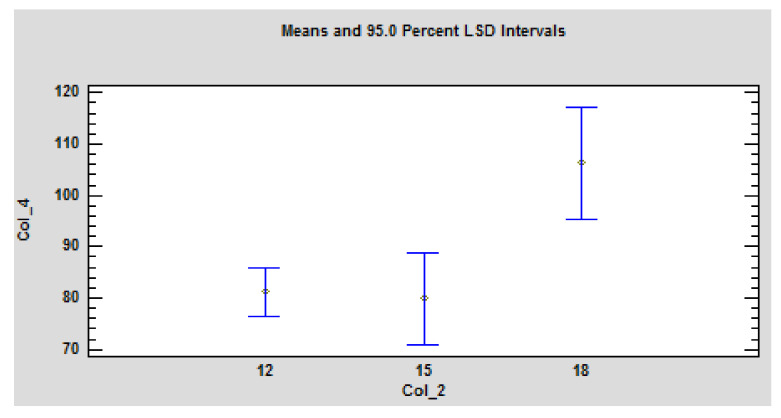
Graphical dependence of the ignition temperature on the thickness of the particleboard. Legend: axis “y”-Col_4 istime to ignition, axis “x”-Col_2 is thickness for heat flux interval <43,48> kW·m^−2^. The values are statistically significant at *p* ≤ 0.05 according to LSD ANOVA.

**Figure 6 polymers-14-01648-f006:**
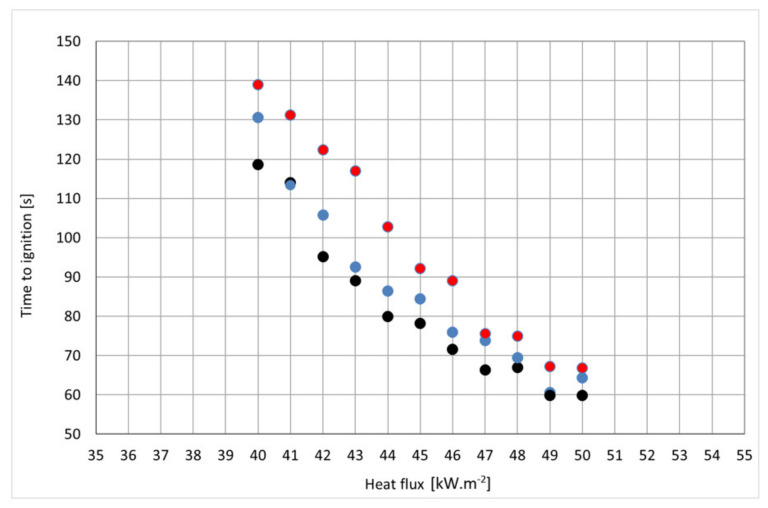
Graphical dependence of the time to ignition on the heat flux and thickness of the particleboard. Legend: black point-12 mm thickness; blue point-15 mm thickness; red point-18 mm thickness.

**Figure 7 polymers-14-01648-f007:**
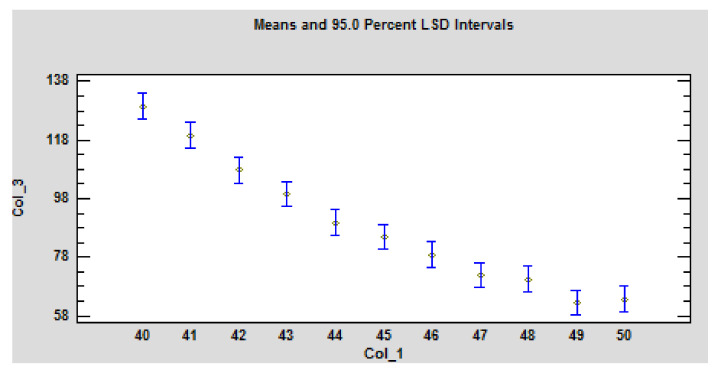
Graphical representation of the statistical evaluation-the influence of the sample thickness and heat flux on the time to ignition under the action of the radiant heat source on the particleboard. Legend: Col_1-heat flux; Col_2-thickness of particleboard samples as the variance of the values shown in blue; Col_3-Time to ignition. The values are statistically significant at *p* ≤ 0.05 according to LSD ANOVA.

**Figure 8 polymers-14-01648-f008:**
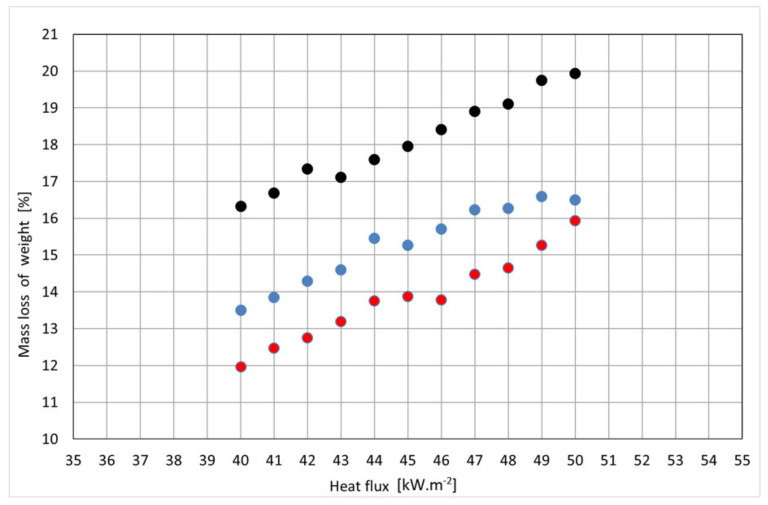
Graphical dependence of average values of weight loss on heat flux and particleboard thickness. Legend: black point-12 mm thickness; blue point-15 mm thickness; red point-18 mm thickness.

**Figure 9 polymers-14-01648-f009:**
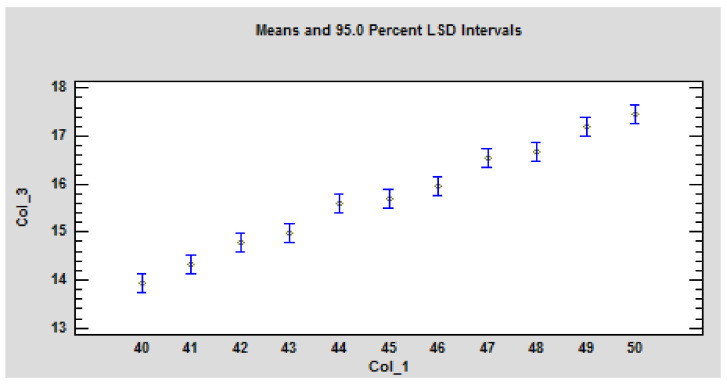
Graphical ANOVA for mass loss (Col_3). Legends: Col_1 is heat flux; Col_2 is thickness of particleboard samples as the variance of the values shown in blue. The values are statistically sig-nificant at *p* ≤ 0.05 according to LSD ANOVA.

**Figure 10 polymers-14-01648-f010:**
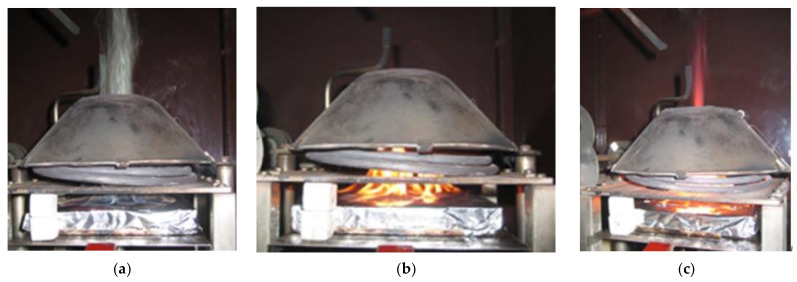
Measurements of time to ignition and weight loss of particleboards with thickness of 12 mm at (**a**–**c**) 40 mm from the ignition source; (**d**–**f**) 60 mm from the ignition source.

**Figure 11 polymers-14-01648-f011:**
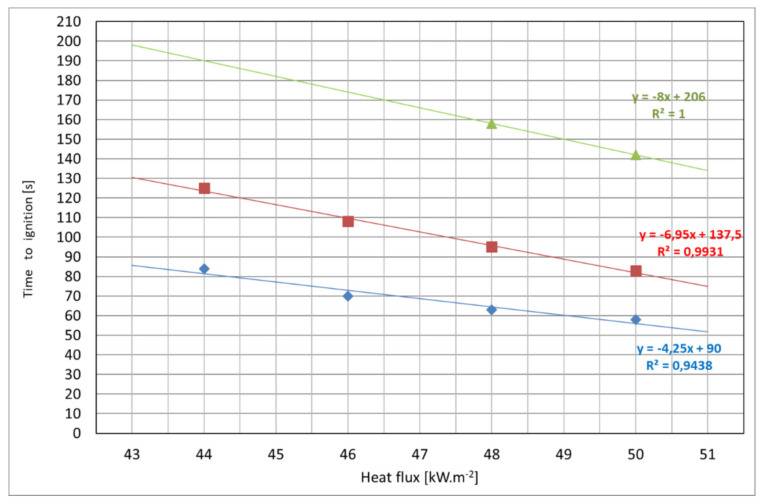
Graphical dependence of the time to ignition on the heat flux (44, 46, 48 and 50 kW·m^−2^) and the distance of the particleboard with thickness of 12 mm from the ignition source. Legend: blue-20 mm; red-40 mm and green-60 mm.

**Figure 12 polymers-14-01648-f012:**
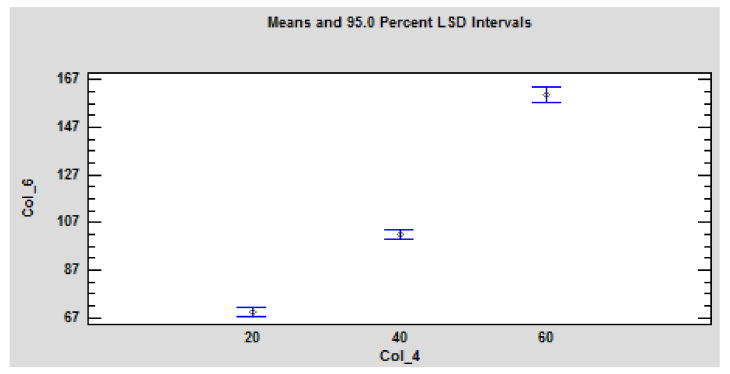
Graphical representation of the statistical evaluation-the influence of the sample position and heat flux on the time to ignition source on the particleboards. Legend: Col_4-position of the sample i.e., distance of the sample from the source; Col_6-time to ignition. The values are statistically significant at *p* ≤ 0.05 according to LSD ANOVA.

**Table 1 polymers-14-01648-t001:** Physical and chemical properties and fire-technical characteristics of particleboards in thicknesses of 12–18 mm [31,32].

Parameters	Thickness of Particleboard Sample (mm)		
	12	15	18
Density (kg·m^−3^) (average)	690	713	644
Moisture (%)	5.05	5.25	5.45
Bending strength (N·mm^−2^)	13.2	12.5	12
Modulus of elasticity (N·mm^−2^)	2500	2450	2750
Swelling after 24 h (%)	3.5	3.5	3.5
Thermal conductivity (W·m^−2^·K^−1^)	0.10–0.14	0.10–0.14	0.10–0.14
Free formaldehyde content (mg·100 g^−1^) (Emission class E1)	6.5	6.5	6.5
Reaction to fire	D-s1, d0		

**Table 2 polymers-14-01648-t002:** Time to ignition and mass loss in samples with different thickness using heat fluxes of 40 to 50 kW·m^−2^ at adistance of 20 mm.

Radiant Heat Flux (kW·m^−2^)	Thickness (mm)	Density (kg·m^−3^)	Thermal Inertia (kJ^2^·m^−4^·s^−1^·K^−2^)	Time to Ignition (s)	Weight Loss (%)
40	12	689 ± 10	0.32 ± 0.002	130.6 ± 3.44	16.3 ± 0.3
	15	711 ± 10	0.31 ± 0.026	118.6 ± 3.83	13.5 ± 0.3
	18	644 ± 6	0.27 ± 0.080	139.0 ± 3.16	11.9 ± 0.2
41	12	689 ± 9	0.32 ± 0.002	114.0 ± 3.52	16.7 ± 0.29
	15	714 ± 11	0.33 ± 0.002	113.4 ± 4.03	13.8 ± 0.33
	18	644 ± 7	0.27 ± 0.082	131.2 ± 2.99	12.5 ± 0.21
42	12	688 ± 9	0.32 ± 0.002	95.2 ± 6.82	17.3 ± 0.39
	15	714 ± 10	0.33 ± 0.002	105.8 ± 3.06	14.3 ± 0.32
	18	645 ± 6	0.31 ± 0.001	122.4 ± 1.96	12.7 ± 0.18
43	12	691 ± 10	0.32 ± 0.002	89.0 ± 5.215	17.1 ± 0.52
	15	716 ± 11	0.33 ± 0.002	92.6 ± 3.441	14.6 ± 0.37
	18	642 ± 7	0.31 ± 0.008	117.0 ± 5.513	13.2 ± 0.17
44	12	691 ± 10	0.32 ± 0.002	80.0 ± 5.37	17.6 ± 0.41
	15	715 ± 10	0.33 ± 0.002	86.4 ± 4.88	15.4 ± 0.35
	18	643 ± 7	0.31 ± 0.002	102.8 ± 4.31	13.7 ± 0.24
45	12	691 ± 9	0.321 ± 0.002	78.2 ± 0.748	17.9 ± 0.30
	15	714 ± 11	0.327 ± 0.002	84.4 ± 2.057	15.3 ± 0.29
	18	645 ± 7	0.311 ± 0.002	92.2 ± 2.481	13.9 ± 0.29
46	12	690 ± 11	0.32 ± 0.002	71.6 ± 1.62	18.4 ± 0.52
	15	711 ± 9	0.33 ± 0.002	76.0 ± 2.28	15.7 ± 0.29
	18	644 ± 8	0.31 ± 0.001	89.0 ± 7.97	13.8 ± 0.56
47	12	690 ± 11	0.32 ± 0.002	66.4 ± 2.87	18.9 ± 0.29
	15	715 ± 10	0.33 ± 0.002	73.8 ± 0.80	16.2 ± 0.36
	18	645 ± 8	0.31 ± 0.002	75.6 ± 3.72	14.5 ± 0.34
48	12	689 ± 10	0.32 ± 0.002	64.4 ± 1.49	19.1 ± 0.34
	15	710 ± 8	0.33 ± 0.001	69.4 ± 1.96	16.3 ± 0.37
	18	644 ± 7	0.31 ± 0.001	75.0 ± 2.00	14.6 ± 0.22
49	12	692 ± 11	0.32 ± 0.002	60.6 ± 2.24	19.7 ± 0.44
	15	713 ± 10	0.33 ± 0.002	66.0 ± 2.28	16.6 ± 0.33
	18	644 ± 11	0.31 ± 0.002	67.2 ± 1.17	15.2 ± 0.13
50	12	689 ± 8	0.32 ± 0.001	59.8 ± 2.64	19.9 ± 0.41
	15	713 ± 11	0.33 ± 0.002	64.4 ± 2.50	16.5 ± 0.33
	18	643 ± 7	0.31 ± 0.001	66.8 ± 2.09	15.9 ± 0.94

**Table 3 polymers-14-01648-t003:** Analysis of variance for Col_4 Time to ignition-type III sums of squares.

Source	Sum of Squares	Df	Mean Square	F-Ratio	*p*-Value
Covariates					
Col_3 Thermal interaction	664.125	1	664.125	1.34	0.2481
Main Effects					
Col_2 Board thickness	4018.06	2	2009.03	4.06	0.0190
Residual	79,573.2	161	494.244		
Total	87,085.1	164			

All F-ratios are based on the residual mean square error.

**Table 4 polymers-14-01648-t004:** Analysis of variance for Col_3 time toignition-type III sums of squares.

Source	Sum of Squares	Df	Mean Square	F-Ratio	*p*-Value
Main Effects					
Col_1 Heat flux	15,376.7	10	1537.67	58.29	0.0000
Col_2 Board thickness	1405.32	2	702.658	26.64	0.0000
Residual	527.564	20	26.3782		
Total	17,309.6	32			

All F-ratios are based on the residual mean square error.

**Table 5 polymers-14-01648-t005:** Analysis of variance for Col_3 (mass loss)-type III sums of squares.

Source	Sum of Squares	Df	Mean Square	F-Ratio	*p*-Value
Main Effects					
Col_1 Heat flux	40.2861	10	4.028	76.04	0.0000
Col_2 Board thickness	103.654	2	51.826	978.28	0.0000
Residual	1.05955	20	0.0529		
Total	144.999	32			

All F-ratios are based on the residual mean square error.

**Table 6 polymers-14-01648-t006:** Analysis of variance for Col_6 (time to ignition)-type III sums of squares depending on sample thickness and distance from the source.

Source	Sum of Squares	Df	Mean Square	F-Ratio	*p*-Value
Main Effects					
Col_4 Distance from the source	47,980.0	2	23,990.0	534.73	0.0000
Col_5 Heat flux	6798.83	3	2266.28	50.51	0.0000
Residual	1974.02	44	44.8642		
Total	52,250.3	49			

All F-ratios are based on the residual mean square error.

## Data Availability

Not applicable.

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
