# Peer review of "Experimental Study of the Influence of Selected Factors on the Particle Board Ignition by Radiant Heat Flux"

_polymers, 2022, doi:10.3390/polym14091648_

Round 1

Reviewer 1 Report

The manuscript entitled "Experimental Study of the Influence of Selected Factors on the Particle Board Ignition by Radiant Heat Flux" deals with the evaluation of the ignition characteristics of raw and modified particleboard samples exposed to heat flux at different intensity.

In my view, the subject of the submitted paper is out of the main aims and scopes of Polymers, since usually this journal publish research works dealing with the preparation, charachterization, physico-chemical aspects abd engineering of POLYMER (also including biopolymers)-based materials.

Therefore, In my opinion, the submitted manuscript is not suitable for publication on Polymers.

Author Response

Thanks to the reviewer for the review and opinion. Our courage to insert the article into the Journal Polymers was based on an offer to public in Special Issue New Challenges in Wood and Wood-Based Materials II.
In our opinion , this topis contains opportunity to present our research results. Particleboards are mixture polymers, which have good properties, but their reaction for heat has negative consequences. And it is topic our research. We believe that topic is interesting in readers.

Reviewer 2 Report

Some improvement necessary, see attached file.

Author Response

Thank you for the helpful comments we have incorporated into the article. We believe that editing the article has made the article better. Our repeated comments are highlight yellow colour in article. 

We are apologized for English mistakes and errors and we tried full text revision. 
The incorporated details are explained in the attached file.

Reviewer 3 Report

The aim of the paper was to monitor the effect of radiant heat on the surface of particleboard according to the modified ISO 5657: 1997 standard. The topic is interesting for practice too. Authors should add some changes and additions in the text.

Authors write "particleboard" together and separately in the text. The word "particleboard" should be the same in all text.

Keywords

  • 1-2 other keywords should be added e.g. thermal resistance, ANOVA

Introduction

  • Authors should add more their own papers. Readers want to know the scientific and research experience of the authors.

Materials and Methods

  • Authors wrote "... Particle boards contain coniferous softwood chips, mainly spruce and urea-formaldehyde adhesive mixture. Large-size wood materials form the largest percentage of wood... " More information are necessary (e.g. properties of the urea-formaldehyde resin and hardener, preparation of the glue mixture).
  • Why Authors selected for the investigations particleboard from BUČINA DDD Company in Zvolen /Slovakia/? Some papers of the scientists whose analyse particleboards properties should be given.

Results and Discussion

  • Results in the Table 2 with the same precision should be given.
  • Points on the Fig. 5 and Fig. 7 with the curves and mathematical model with R2 coefficient should be described (like on the Fig. 10).

Conclusions

  • Conclusions corresponds with the results and should be numbered.

I recommend the paper for the publishing after minor changes and additions.

Author Response

Thank you for the helpful comments we have incorporated into the article. We believe that editing the article has made the article better. Our repeated comments are highlight green colour in article. 

We are apologized for English mistakes and errors and we tried full text revision. 
The incorporated details are explained in the attached file.

We added our actual references:

Marková, I.; Hroncova, E.; Tomaskin, J.; Turekova, I. Thermal analysis of granulometry selected wood dust particles. BioResources 2018, 13, 4, 8041-8060.

Szabová, Z.; Pastier, M.; Harangózo, J.; Chrebet, T. Determination of characteristics predicting the ignition of organic dusts. In Occupational Safety and Hygiene II : 10th Annual Congress of the Portuguese Society of Occupational Safety and Hygiene on Occupational Safety an Hygiene (SPOSHO) Guimaraes, Portugal, 13 - 14 February 2014. Boca Raton : CRC Press, 2014, s. 143-145. ISBN 978-1-315-77352-0.

Turekova, I.; Markova, I. Ignition of Deposited Wood Dust Layer by Selected Sources. Appl. Sci. 2020, 10, 17, 5779.

Vandličková, M., Markova, I., Osvaldová, L. M., Gašpercová, S., and Svetlík, J. Evaluation of African padauk (Pterocarpus soyauxii) explosion dust. BioRes. 2020, 15, 1, 401-414.

We are very sorry, but we cannot complete the request regarding the addition of properties and the preparation of chipboards. We received the researched chipboards as a finished product from a specific manufacturer BUČINA DDD COMPANY in Zvolen. The research was part of the applied research of the behavior of chipboards at different forms of heat for the manufacturer himself.

We added followed references:

Kristak, L.; Kubovsky, I.; Reh, R. New Challenges in Wood and Wood-Based Materials. Polymers 2021, 13, 15, 2538

Bekhta, P.; Noshchenko, G. ; Reh.,R. ; Kristak, L.; Sedliacik, J. ; Antov, P.; Mirski, R.; Savov, V. Properties of Eco-Friendly Particleboards Bonded with Lignosulfonate-Urea-Formaldehyde Adhesives and pMDI as a Crosslinker. Materials 2021, 14, 17, 4875

Krist'ak, L. Reh, R. Application of Wood Composites. Appl. Sci. 2021, 11,8, 3479

Pedzik, M.; Auriga, R.; Kristak, L.; Antov, P.; Rogozinski, T. Physical and Mechanical Properties of Particleboard Produced with Addition of Walnut (Juglans regia L.) Wood Residues. Materials 2022, 15, 1280.

Baskaran , M.; Nur Adilah Che Hassan Azmi; Hashim , R.; Sulaiman, O. Addition of Urea Formaldehyde Made from Oil Palm Trunk Waste. Journal of Physical Science 2017, 28, 3, 151–159.

Chuan Li Lee, Ch. L.; Chin, K.L.; H'ng P.S.; Chuah, A.L.; Khoo P.S. Enhanced properties of single-layer particleboard made from oil palm empty fruit bunch fibre with additional water-soluble additives. BioRes. 2021,16, 3, 6159-6173.

Mirindi, D., Onchiri, R.O.; Thuo, J. Physico-Mechanical Properties of Particleboards Produced from Macadamia Nutshell and Gum Arabic. Appl. Sci. 2021, 11, 11138

Bardak, S.; Nemli, G.; Bardak, T. The quality comparison of particleboards produced from heartwood and sapwood of European larch. Maderas. Ciencia y tecnología 2019, 21, 4, 511 - 520, 2019

Conclusions are numbered.

Round 2

Reviewer 1 Report

Thank to the Authors for their kind answer. Unfortunately, I’m still convinced that a moreappropriate journal would be suitable for the submitted work. I agree with the Authors when they claim the polymeric origin of particleboards, but in the study no characterisation or discussion involving the kind and the behaviour of the polymeric components are included.

Author Response

Thanks to the reviewer for the review and opinion.

We are very sorry that the reviewer does not agree with the presentation of our results. On the one hand, we agree with the opponent's opinion, on the other hand, we offer research on the behavior of particleboard, where their thermal degradation occurs. It would be interesting to observe changes in the chemical structure after thermal degradation, but this is already the research of our other colleagues.

Nevertheless, we think that the article could be published in the special issue and we ask the reviewer for support.

Reviewer 2 Report

See attached file.

Author Response

We thank the reviewer for valuable advices, comments were made directly in the text and highlighted in blue, which contributed to improvement of the quality of the article. 

Our answers for comments are in attached.
